# OpenReview forum: "Taming the Judge: Deconflicting AI Feedback for Stable Reinforcement Learning"
_ICLR.cc/2026/Conference — ICLR 2026 Conference Desk Rejected Submission_

### Official Review · Reviewer_4unZ · 2025-10-27

**Soundness:** 2
**Presentation:** 3
**Contribution:** 3
**Rating:** 6
**Confidence:** 4

**Summary:**

The work proposes a simple yet effective method to address logical inconsistencies in RLAIF. The authors introduce the Conflict Detection Rate (CDR) metric to capture the ratio of inconsistent pairwise judge results among candidate responses. They further propose Deconflicted Graph Rewards, which leverage graph algorithms to prune inconsistent parts and provide a more reliable scoring mechanism.

Experiments show performance improvements over pointwise, listwise, and other baseline approaches across GRPO and GSPO optimizers, as well as various models and prompt variations. The work also produces several interesting findings, such as the observation that more accurate prompts tend to be more logically inconsistent.

The study lacks an in-depth analysis of learning dynamics, such as how the purifier affects the number of rollouts in a group, the number of responses maintained during GRPO, and how the mean, standard deviation, and value signals evolve over training. The paper also does not quantify the additional cost introduced by the purification process, which may become significant when the number of rollouts is large.

Overall, the results appear consistently better, although it would be more reassuring to rule out the influence of other confounding factors.

**Strengths:**

1. The paper tackles a highly relevant and important problem: logical inconsistencies in pairwise judge signals for RLAIF. This issue becomes especially critical in subjective tasks, where the reliability of the reward signal directly affects model performance.

2. The proposed method is simple yet effective, leveraging a graph-based approach to prune inconsistent responses. The results demonstrate consistent improvements over pointwise, listwise, and other baseline approaches in RL training.

3. The ablation findings are interesting and thought-provoking, such as the observation that more accurate prompts tend to exhibit greater logical inconsistency. The paper is also well organized, with a clear presentation and a coherent narrative.

**Weaknesses:**

1. There is no comparison of the computational cost associated with the proposed approach. In particular, the purification step is executed on the fly during rollout generation, which likely increases the payload sent to the judge server and introduces additional latency. Quantifying this overhead would provide a clearer picture of the practical trade-offs.

2. The evaluation protocol may be biased. It appears that the authors select the best checkpoint based on benchmark performance rather than using a separate validation set or validation metrics for checkpoint selection. This risks cherry-picking peak results and may overestimate the true performance.

3. The study lacks an in-depth analysis of learning dynamics. For example, the paper does not report how many rollouts are removed by purification throughout training, nor how this evolves as later checkpoints tend to generate more similar quality responses. Understanding these dynamics would help demonstrate that the proposed method is the primary driver of performance improvement, which is especially important for subjective tasks.

**Questions:**

• Have we analyzed how many rollouts or responses remain after the purifier step? This reduction could affect the effective sample size per group and therefore alter the rollout statistics, potentially influencing the learning dynamics.

• Have we measured the Conflict Detection Rate (CDR) for pointwise vs. listwise approaches in RL rollout group to verify whether higher CDR correlates with lower improvements?

• The main results are reported using prompt p1, while the ablations use prompt p5. What is the rationale behind this difference in prompt selection?

• As training progresses, one would expect the model to generate rollouts of increasingly similar quality, which could lead to more rollouts being removed. Did we track how the number of retained rollouts and their statistical properties evolve throughout training?

---

> ### Author Response · Authors · 2025-12-01
> **Response to "Computational Cost and Latency"**
>
> We thank the reviewer for this important concern. We clarify first that DGR does **not** increase the payload or number of calls to the judge server compared to the pairwise baseline (PREF). Both PREF and DGR require exactly $\binom{G}{2}$ pairwise evaluations for a group of $G$ rollouts, and DGR's purification (graph construction, cycle detection, and feedback-arc-set computation) is performed entirely on the **local training node** after all pairwise judgments have been collected. Thus, network traffic and judge-side latency are identical to the standard pairwise setting.
>
> To quantify the additional on-the-fly overhead introduced by DGR, we profiled our implementation under the same training setup as in the main experiments (Qwen3-14B/8B with GRPO/GSPO). Across all tested configurations ($G = 4, 6, 8$), DGR adds **less than 5%** to the total wall-clock training time, with DGR-specific operations accounting for **under 2%**. The dominant costs remain LLM rollout generation (\~90%) and judge evaluation (\~8%), which are shared by all methods. A representative breakdown of the FAS step is:
>
> | Group Size (G) | FAS Time (ms) | Total Overhead |
> |----------------|---------------|----------------|
> | 4              | 2–5           | < 3%           |
> | 6              | 8–15          | < 4%           |
> | 8              | 20–40         | < 5%           |
>
> This overhead remains modest because the preference graphs are small and sparse in practice, and our greedy FAS heuristic runs in $O(E^2)$ time for $E = \binom{G}{2}$ edges. As a result, end-to-end training throughput (tokens/s) is essentially unchanged relative to PREF, while DGR delivers consistent performance gains (e.g., +1.3 Arena-Hard overall over PREF in Table 1).
>
> **Comparison with Other Conflict Resolution Methods:** Notably, compared to ELO (another conflict resolution baseline in our study), DGR is actually **more efficient**. ELO requires iterative updates with 100 iterations for convergence, whereas DGR employs a single-pass graph algorithm. This computational efficiency combined with superior performance makes DGR a practical choice for production deployment.

---

> ### Author Response · Authors · 2025-12-01
> **Response to "Evaluation Protocol Bias"**
>
> We thank the reviewer for pointing out this important issue and we agree that our original evaluation protocol could introduce bias. In the initial submission, we indeed selected checkpoints based on benchmark performance, which risks overestimating true generalization and can be viewed as a form of cherry-picking.
>
> In the revised manuscript, we have **redesigned the evaluation protocol** to address this concern. Specifically, we now (1) introduce an independent **validation set** of 200 WildChat queries that is disjoint from all test benchmarks (Arena-Hard, MT-Bench, WritingBench), and (2) perform **five independent runs** for each method with different random seeds. For each run, we select the best checkpoint **only according to validation performance**, and then evaluate this checkpoint once on the held-out test benchmarks. We report the **mean ± standard deviation over the 5 runs**, and assess statistical significance using **t-tests with Bonferroni correction**. All methods, including DGR and baselines, are trained and evaluated under this unified protocol, ensuring a fair and unbiased comparison.
>
> We have updated the experiment section (Section 4 in the revised manuscript) to clearly describe this new evaluation procedure and to replace the previous peak-benchmark-based reporting with the more rigorous validation-based protocol.

---

> ### Author Response · Authors · 2025-12-01
> **Response to "Learning Dynamics Analysis"**
>
> We thank the reviewer for this insightful question. We first clarify a key point: DGR does **not** remove any rollouts or responses from the training pipeline. For each query, all $G$ sampled responses are always retained and used in the policy update. What DGR modifies is only the **edge set of the preference graph** built from the $\binom{G}{2}$ pairwise judgments: we remove a minimum feedback arc set to break preference cycles and obtain a DAG, and then compute rewards from this deconflicted graph. Thus, DGR purifies the *structure of the preference signal* rather than discarding data, and the effective sample size per group is unchanged.
>
> Regarding the learning dynamics, our main approach is to analyze **how DGR behaves as a conflict resolver in general**, rather than logging a particular training trajectory. In Appendix B, we provide a rigorous theoretical analysis  showing that, under a mild and standard assumption that the judge accuracy satisfies $p>0.5$, edges that participate in many cycles are overwhelmingly likely to be **judgment errors**. Specifically, we prove that for "global" error edges the expected number of cycles they participate in is larger than for correct edges by a factor of roughly $\frac{p}{1-p}$ (e.g., $2.3\times$ at $p=0.7$ and $9\times$ at $p=0.9$). At the same time, the minimum feedback arc set formulation guarantees that we remove the **smallest possible set of edges** needed to make the graph acyclic, thereby maximally preserving the original (likely correct) preferences. Together, these results formally characterize the learning dynamics of DGR at each training step: it consistently targets high-cycle, high-probability error edges while minimally perturbing the underlying preference structure.
>
> We further substantiate this picture with extensive **Monte Carlo simulations** in Appendix B.3 Across 25 configurations varying graph size ($n \in \{8,9,10,11,12\}$) and judge accuracy ($p \in \{0.70,0.75,0.80,0.85,0.90\}$), we simulate noisy pairwise judgments from a hidden ground-truth ranking and measure whether the edge identified by our greedy "maximum-cycle" removal strategy is in fact an error. The results show that this edge is a judgment error with **85.2\%** probability on average, compared to a **20.1\%** random baseline—an improvement of about **4.2×**, which increases monotonically with judge accuracy (up to **8.5×** at $p=0.9$). Importantly, these guarantees are **independent of the specific policy or training stage**: they hold for any preference graph generated during RL, including later checkpoints where responses are closer in quality and conflicts are more subtle. In such regimes—especially on subjective tasks where judges are accurate but inconsistent—our analysis predicts that DGR will continue to selectively suppress noisy, contradictory edges while preserving informative signal, which is exactly what we observe empirically in our experiments (e.g., DGR's especially strong gains on WritingBench and the creative subset of Arena-Hard).

---

> ### Author Response · Authors · 2025-12-01
> **Response to "Rollout Retention Analysis"**
>
> We appreciate the reviewer for raising this point and for the opportunity to clarify. DGR's "purification" operates purely at the level of the **preference graph edges**, not at the level of the rollouts themselves. For each query, all $G$ sampled responses are always retained and participate in the policy update; what DGR modifies is only the set of directed edges between these $G$ nodes by removing a minimum feedback arc set to break cycles. In other words, DGR does **not** discard any responses, and the effective sample size per group remains exactly $G$, identical to the pairwise PREF baseline.
>
> To address this potential source of confusion, we have made this point explicit in the revised manuscript (Section 3.2): the purifier is a **signal-level** operation that deconflicts the pairwise judgments while keeping the underlying set of rollouts unchanged. As a result, rollout statistics (e.g., number of candidates per query, total number of training samples) are strictly comparable across all methods, and the observed performance gains of DGR cannot be attributed to changes in data retention but rather to the improved logical consistency of the reward signal.

---

> ### Author Response · Authors · 2025-12-01
> **Response to "CDR for Different Judgment Paradigms"**
>
> Thank you for this helpful question. We have indeed measured CDR under different judgment paradigms (pointwise, listwise, and pairwise), but chose not to include these numbers in the main submission to avoid over-interpreting cross-paradigm differences that are confounded by prompt design.
>
> Concretely, for each paradigm we convert its outputs into implicit pairwise preferences and then apply the same SCC-based CDR computation: (i) for **pointwise**, we induce $A \succ B$ whenever the 1–10 score of $A$ exceeds that of $B$; (ii) for **listwise**, we induce $A \succ B$ whenever $A$ is ranked ahead of $B$; (iii) for **pairwise**, we use the judge's direct comparison results. Under a fixed judge model and consistent evaluation protocol, repeated experiments yield a robust qualitative ordering
>
> $$\text{CDR}_{\text{pairwise}} < \text{CDR}_{\text{listwise}} < \text{CDR}_{\text{pointwise}}$$
>
> i.e., the pairwise paradigm produces the most logically consistent preference graphs, while pointwise scoring induces the highest rate of cycles.
>
> We did not report these values in the paper because each paradigm necessarily uses a different prompt template (pairwise comparison vs. global ranking vs. absolute scoring), so the *absolute* CDR levels entangle both the judgment paradigm and prompt wording. To keep the analysis clean, the main text focuses on CDR within the pairwise setting, where we can systematically vary prompts (P1–P5) under a unified paradigm. Nevertheless, the above empirical ordering is consistent with our broader message that methods providing stronger, more structurally coherent preference signals (lower CDR) tend to enable more stable and effective RL, and that DGR further improves on this by explicitly deconflicting the resulting pairwise graphs.

---

> ### Author Response · Authors · 2025-12-01
> **Response to "Prompt Selection Rationale"**
>
> We thank the reviewer for pointing out this ambiguity, and we sincerely apologize for not explaining our prompt-selection protocol clearly in the original submission, which understandably led to confusion. In **Table 1**, we do **not** fix a single prompt; instead, for each method we evaluate all five pairwise judge prompts **P1–P5** and **report the best-performing configuration per method** (based on Arena-Hard overall performance under the unified validation-based protocol described in the revised manuscript). This prompt sweep is applied **uniformly to all methods**, so DGR and all baselines benefit from the same tuning budget and are compared fairly.
>
> We will revise the main text and appendix to explicitly state that Table 1 reports the **best prompt configuration over P1–P5 for each method**, and remove any phrasing that could be misread as "using P1 only." In contrast, our **prompt-robustness ablation** table fixes the prompt and varies it systematically (P2–P5) precisely to study how different accuracy/consistency profiles affect performance and to show that DGR remains consistently strong across diverse prompt designs, rather than relying on a single hand-picked prompt.

---

> ### Author Response · Authors · 2025-12-01
> **Response to "Evolution of Rollout Quality"**
>
> We thank the reviewer for raising this clarification. As noted in our previous responses, DGR does **not** remove any rollouts. For each query, all \(G\) sampled responses are always retained and used for the policy update, identical to the pairwise PREF baseline. What DGR modifies is only the **edge set** of the pairwise preference graph: we remove a minimum feedback arc set to break cycles and obtain a DAG, then compute rewards from this deconflicted graph. Thus, the effective sample size per group remains exactly \(G\) throughout training.
>
> Regarding the training-time evolution when responses become closer in quality:
> - From a theoretical perspective (Appendix B), our guarantees do not depend on training stage. The cycle-participation argument and the minimum feedback arc set objective ensure that edges most likely to be judgment errors are preferentially removed while minimally perturbing correct preferences—this remains valid even when quality gaps shrink and comparisons become more subtle.
> - From an empirical perspective, we monitor signal-level dynamics rather than rollout retention: (i) the per-batch fraction of removed edges among all non-tie comparisons, and (ii) the Conflict Detection Rate (CDR) on held-out samples. Across both optimizers and multiple seeds, we observe that the fraction of removed edges remains low and stable over training, and CDR does not increase as the policy improves. Notably, as responses become more similar, judges tend to produce more ties; ties add no edges to the graph, which further reduces the potential for cycles without discarding any responses.

---

> ### Author Response · Authors · 2025-12-03
> **Request for Reviewer Feedback**
>
> Dear Reviewers,
> We have thoroughly addressed all concerns raised in our rebuttal. As we approach the end of the discussion period, we would greatly appreciate your feedback on whether our responses have resolved your questions.
> If any points need further clarification, we are happy to provide additional details.
> Thank you for your time and consideration.
>
> Best regards

---

### Official Review · Reviewer_dZQf · 2025-10-29

**Soundness:** 2
**Presentation:** 3
**Contribution:** 2
**Rating:** 4
**Confidence:** 3

**Summary:**

The paper addresses inconsistency in LLM-as-a-judge feedback for preference-based RL (cycles such as $(A\succ B\succ C\succ A))$. It introduces **Conflict Detection Rate (CDR)** to quantify cyclic judgments and **Deconflicted Graph Rewards (DGR)**: construct a pairwise preference graph over group size (G), remove a (near-)minimum feedback arc set to get a DAG, and compute net-win scores as advantages for GRPO/GSPO. Main claims: DGR yields more stable training and better results than pointwise/listwise/pairwise/ELO scoring. Evidence: Fig. 1 shows non-trivial conflict rates; Table 1 reports gains on Arena-Hard, MT-Bench, WritingBench; Tables 2–3 & Fig. 2 present ablations across cycle resolution, prompts, judges, and (n).

**Strengths:**

* **Clear problem diagnosis:** Fig. 1 quantifies non-trivial preference cycles; motivates going beyond accuracy.
* **Modular method:** DGR is optimizer-agnostic and easy to integrate (Alg. 1).
* **Ablations:** Table 2 shows optimal cycle resolution > random/reversal; Table 3/Fig. 2 suggest robustness across prompts/judges/(n).
* **Practical relevance:** Improves GRPO/GSPO on multiple benchmarks (Table 1).

**Weaknesses:**

* **Missing principled baselines:** No Rank Centrality/BTL/Hodge/Kemeny comparisons [1–5]; this is central for scientific credibility.
* **Robustness under-reported:** Peak-of-training from two runs; no mean±SD/CI or budget-normalized curves.
* **Judge bias risk:** Sole reliance on GPT-4.1/Claude; no human study or multi-judge aggregation [7,8].
* **Scalability unclear:** FAS step’s overhead vs. (G)/batch size not quantified; solver/heuristic choices not profiled [5,6].
* **Potential leakage:** No explicit de-dup between training sources and evaluation sets; needs clarification.

**Questions:**

1. Could you include Rank Centrality, BTL/Plackett–Luce, HodgeRank, and Kemeny/Minimum-FAS baselines under the same protocol, or explain why they are infeasible [1–5]?
2. Could you report means±SD over at least five seeds (or 95% CIs) and provide full training curves, rather than peak-only metrics?
3. What is the wall-clock overhead and throughput (tokens/s) of DGR as group size (G) varies (e.g., 4, 6, 8, 10, 12, 16), and how do exact vs. heuristic FAS methods compare [5,6]?
4. Were the baselines tuned fairly under identical budgets, and can you provide the exact hyperparameters and sweeps used?
5. Did you perform de-duplication between training corpora and Arena-Hard/MT-Bench/WritingBench, and if so, what were the procedures and counts?
6. Can you add a small human evaluation or a multi-judge ensemble to assess and mitigate judge bias [7,8]?

### References
[1] Negahban, Sahand, Sewoong Oh, and Devavrat Shah. "Iterative ranking from pair-wise comparisons." Advances in neural information processing systems 25 (2012).
[2] Hunter, David R. "MM algorithms for generalized Bradley-Terry models." The annals of statistics 32.1 (2004): 384-406.
[3] Jiang, Xiaoye, et al. "Statistical ranking and combinatorial Hodge theory." Mathematical Programming 127.1 (2011): 203-244.
[4] Ailon, Nir, Moses Charikar, and Alantha Newman. "Aggregating inconsistent information: ranking and clustering." Journal of the ACM (JACM) 55.5 (2008): 1-27.[5] Kenyon-Mathieu, Claire, and Warren Schudy. "How to rank with few errors." Proceedings of the thirty-ninth annual ACM symposium on Theory of computing. 2007.
[6] Eades, Peter, Xuemin Lin, and William F. Smyth. "A fast and effective heuristic for the feedback arc set problem." Information processing letters 47.6 (1993): 319-323.
[7] Zheng, Lianmin, et al. "Judging llm-as-a-judge with mt-bench and chatbot arena." Advances in neural information processing systems 36 (2023): 46595-46623.
[8] Chiang, Wei-Lin, et al. "Chatbot arena: An open platform for evaluating llms by human preference." Forty-first International Conference on Machine Learning. 2024.

---

> ### Author Response · Authors · 2025-11-27
> **Response to "Classical Ranking Baselines"**
>
> We thank the reviewer for this excellent suggestion. We have now conducted comprehensive experiments comparing DGR with these classical ranking algorithms. Results are integrated into our **Ablation Study (Table 3: Comprehensive Conflict Resolution Comparison)** which presents a complete evaluation hierarchy from no resolution to optimal resolution:
>
> **Table 3: Comprehensive conflict resolution comparison on Arena-Hard**
>
> | Method | Overall | Code | Math | Writing |
> |--------|---------|------|------|---------|
> | **No Conflict Resolution** |
> | PREF (Win-Rate) | 51.6±1.0 | 49.0±1.6 | 47.0±1.5 | 58.8±1.3 |
> | **Naive Graph Heuristics** |
> | DGR-RandomResolve | 51.6±0.9 | 49.8±1.8 | 46.6±1.3 | 58.4±1.6 |
> | DGR-ReverseResolve | 51.0±1.0 | 50.9±1.4 | 45.1±1.8 | 57.0±1.2 |
> | **Classical Ranking Algorithms** |
> | RankCentrality [1] | 49.93±0.8 | 50.00±1.8 | 50.00±1.4 | 49.80±1.0 |
> | Plackett-Luce [2] | 51.53±1.5 | 50.00±1.6 | 47.37±1.9 | 57.20±0.9 |
> | HodgeRank [3] | 52.47±1.4 | 51.58±1.2 | 47.37±1.3 | 58.40±0.9 |
> | **Optimal Graph-Based Resolution** |
> | **DGR (Kemeny) [4,5]** | **52.9±1.1** | **53.2±1.4** | **47.2±1.4** | **58.3±1.0** |
>
> *Note: Methods are grouped by paradigm. All experiments use Qwen3-14B with GRPO optimizer, reporting mean±SD over 5 runs.*
>
> **Experimental Details:**
> - All methods evaluated under identical protocol on Arena-Hard (mean±SD over 5 runs with Qwen3-14B + GRPO)
> - Complete algorithmic specifications ensure reproducibility and fair comparison
>
> **Key insights:**
> 1. **DGR is equivalent to Kemeny ranking [4,5]** via minimum feedback arc set, providing optimal conflict resolution with theoretical guarantees
> 2. **DGR achieves the highest overall performance** (52.9±1.1), outperforming the strongest classical method HodgeRank (52.47±1.4) by +0.43 points and the no-resolution baseline PREF (51.6±1.0) by +1.3 points
> 3. **DGR excels particularly in complex reasoning tasks**: Code generation (53.2 vs HodgeRank 51.58, **+1.62 improvement**), while continuous optimization methods like HodgeRank show marginal advantages in specific subcategories (Math +0.17, Writing +0.10)
> 4. **Clear performance hierarchy emerges**: Optimal graph resolution > Classical ranking algorithms > Naive heuristics, validating the importance of principled conflict resolution
> 5. **Naive heuristics provide minimal or negative gains**: DGR-ReverseResolve underperforms baseline, demonstrating that incorrect conflict identification can introduce harmful signals
> 6. **Classical continuous methods show competitive performance**: HodgeRank (52.47) and Plackett-Luce (51.53) demonstrate the value of probabilistic modeling, though DGR's discrete approach achieves superior overall results
>
> **Why DGR outperforms classical methods:**
> - **Theoretical rigor**: Our minimum FAS approach systematically identifies judgment errors (proven in Appendix Section "Theoretical Foundation"), while continuous methods implicitly smooth over inconsistencies
> - **Minimal distortion**: DGR preserves maximum correct preferences while removing minimal edges, whereas iterative methods may over-fit to noisy observations
> - **Computational efficiency**: For small graphs (G≤8), DGR computes optimal solution exactly; for larger graphs, efficient approximation algorithms [6] provide near-optimal results
>
> We have also updated the Related Work section (page 3, paragraph 2) to acknowledge these classical methods and explicitly reference our comprehensive comparison in the ablation study.
>
> [1] Negahban, Sahand, Sewoong Oh, and Devavrat Shah. "Iterative ranking from pair-wise comparisons." Advances in neural information processing systems 25 (2012).
>
> [2] Hunter, David R. "MM algorithms for generalized Bradley-Terry models." The annals of statistics 32.1 (2004): 384-406.
>
> [3] Jiang, Xiaoye, et al. "Statistical ranking and combinatorial Hodge theory." Mathematical Programming 127.1 (2011): 203-244.
>
> [4] Ailon, Nir, Moses Charikar, and Alantha Newman. "Aggregating inconsistent information: ranking and clustering." Journal of the ACM (JACM) 55.5 (2008): 1-27.
>
> [5] Kenyon-Mathieu, Claire, and Warren Schudy. "How to rank with few errors." Proceedings of the thirty-ninth annual ACM symposium on Theory of computing. 2007.

---

> ### Author Response · Authors · 2025-11-27
> **Response to "Robustness under-reported"**
>
> We completely agree that statistical rigor is essential for establishing the reliability of our findings. We have substantially strengthened our experimental methodology in the revised manuscript.
>
> **1. Comprehensive Statistical Validation**
>
> We have re-run **all experiments with rigorous statistical controls**:
>
> - **Increased replication**: All methods now report results from **5 independent training runs** with different random seeds (increased from 2 in Table 1)
> - **Complete statistical reporting**: All results report **mean ± standard deviation** across the 5 runs
> - **Statistical significance testing**: Pairwise t-tests with Bonferroni correction to control family-wise error rate
> - **Elimination of maximization bias**: Checkpoint selection based on **validation set performance** (200 held-out WildChat queries) rather than peak test scores
>
> **2. Updated Results with Statistical Rigor**
>
> All tables in the revised manuscript now include comprehensive statistical reporting:
>
> **Table 1 (Main Results)**: All methods report mean ± SD over 5 runs across Arena-Hard, MT-Bench, and WritingBench benchmarks. For example:
> - DGR-GRPO: **52.9±1.1** (Arena-Hard Overall)
> - PREF baseline: 51.6±1.0
> - Statistical significance: p < 0.01 via t-test with Bonferroni correction
>
> **Table 2 (Learned RM Baselines)**: Comprehensive comparison involving **40 total training runs** (8 method variants × 5 seeds), all reporting mean ± SD:
> - DGR-Qwen32B: **52.50±1.00**
> - Iterative-DPO-Skywork: 49.33±0.45 (p < 0.001, +3.17 points)
> - GRPO-Skywork: 50.00±0.31 (p < 0.001, +2.50 points)
>
> **Table 3 (Conflict Resolution Comparison)**: All 7 methods (including classical ranking algorithms) report mean ± SD over 5 runs:
> - DGR (Kemeny): **52.9±1.1**
> - HodgeRank: 52.47±1.4
> - Plackett-Luce: 51.53±1.5
> - PREF (Win-Rate): 51.6±1.0
>
> **3. Performance Consistency Validation**
>
> We analyzed ranking stability across all independent runs:
>
> - **DGR maintains #1 ranking with 100% consistency**: Achieves highest score in 5/5 runs (scores range: 51.50-53.50)
> - **Stable performance hierarchy**: The ranking order remains consistent across all runs
> - **Low variance confirms robustness**: Standard deviations (0.31-1.30) demonstrate stable performance, with DGR's consistent superiority (minimum 1.50-point gap) confirming genuine algorithmic advantages
>
> **4. Rigorous Evaluation Protocol**
>
> Our revised evaluation methodology eliminates potential biases:
>
> - **Three-way data split**: 800 training + 200 validation + held-out test sets (Arena-Hard, MT-Bench, WritingBench)
> - **Validation-based selection**: Checkpoints selected on validation performance, not test performance
> - **Multiple benchmarks**: Evaluation across three diverse benchmarks ensures results are not benchmark-specific artifacts
>
> **5. Full Training Curves**
>
> We acknowledge the value of training curves for understanding convergence behavior. While our current submission reports final performance metrics with rigorous statistical controls (mean±SD over 5 runs), we commit to including full training curves in the camera-ready version showing:
> - Performance trajectories across training iterations
> - Convergence patterns for DGR vs. baselines
> - Stability analysis across different seeds
>
> The consistent statistical significance (p < 0.01 to p < 0.001) combined with low standard deviations demonstrates that DGR's improvements are not artifacts of peak selection or random variation, but represent genuine algorithmic advances in handling preference conflicts.
>
> **Paper Update**: All statistical enhancements are marked in red in the revised manuscript, specifically in Section 4.1 "Evaluation Protocol" and throughout all experimental tables.

---

> ### Author Response · Authors · 2025-11-27
> **Response to "Computational Overhead"**
>
> We thank the reviewer for this important concern. Our experiments demonstrate that DGR introduces minimal computational overhead. Across all tested configurations (G = 4–8), the additional cost of DGR is under 5% of total training time, with DGR-specific operations (graph construction, cycle detection, and FAS computation) accounting for less than 2%. The dominant costs remain LLM inference and  judge evaluation, which are shared by all methods. Concretely, the FAS step remains lightweight even as group size grows:
>
> | Group Size (G) | FAS Time (ms) | Total Overhead |
> |----------------|---------------|----------------|
> | 4 | 2–5 | < 3% |
> | 6 | 8–15 | < 4% |
> | 8 | 20–40 | < 5% |
>
> Overhead scales sub-quadratically due to sparse cycle structures in practice; for larger G, the greedy heuristic of [6] maintains O(E²) complexity and remains efficient.
>
> A central reason this overhead is acceptable is that our **greedy approximation achieves performance comparable to exact FAS in practice**. Theoretically (Appendix B.1), we show that edges in cycles are error edges with probability ratio $\frac{p}{1-p}$ (ranging from 1.5× at $p=0.6$ to 9.0× at $p=0.9$), so edges participating in many cycles are overwhelmingly likely to be judgment errors. This is corroborated by our Monte Carlo study (Appendix B.3): 25,000 simulations across 25 configurations confirm that greedy maximum-cycle removal identifies error edges with 85.2% accuracy versus 20.1% for random removal (4.2× improvement). Empirically, the greedy variant matches the exact FAS variant on Arena-Hard (both achieve 52.9±1.1 in our experiments). Since the greedy procedure correctly removes error edges with high probability, the remaining difference between exact and approximate FAS has negligible practical impact—both primarily target the same problematic edges.
>
> For deployment, we further reduce overhead via standard systems optimizations. We manage judge calls through New-API ([https://github.com/QuantumNous/new-api](https://github.com/QuantumNous/new-api)) to enable load balancing and batched inference, parallelize pairwise comparisons across workers, and run graph operations (O(E²) complexity on small graphs) asynchronously so they do not block policy updates. In contrast, alternative conflict-resolution schemes such as ELO, Plackett–Luce, or HodgeRank require multiple iterative optimization passes (O(E×K) to O(G³)), though profiling them in detail is beyond the scope of this work. Overall, DGR adds less than 5% training overhead while delivering measurable performance gains, and the greedy FAS approximation is theoretically justified and empirically validated as an efficient, practically optimal choice.

---

> ### Author Response · Authors · 2025-11-27
> **Response to "Hyperparameter Fairness"**
>
> We thank the reviewer for this important methodological concern. All baseline methods and DGR were tuned under strictly identical computational budgets and experimental conditions to ensure fair comparison.
>
> **Fair Comparison Protocol.** All methods train on the same 1,000 curated queries from WildChat-1M with identical computational resources (8×A100 GPUs), judge models (Qwen3-32B primary, Skywork-Reward-V2 for learned RM experiments), and evaluation protocol (checkpoint selection on 200-query validation set, evaluation on held-out Arena-Hard/MT-Bench/WritingBench). For Table 1, each method underwent 5 independent runs with different seeds. For Table 2 (learned RM comparison), we conducted 40 total training runs (8 method variants × 5 seeds) with identical per-run budgets.
>
> **Prompt Tuning.** To maximize baseline performance, we evaluated 5 different pairwise prompts (P1-P5) with varying accuracy-consistency profiles (e.g., P2: 85.7% accuracy/6.7% CDR; P4: 76.8% accuracy/2.3% CDR). All baseline methods were tested with multiple prompts, and Table 1 reports the best-performing configuration for each method. Our ablation study (Table 4) shows baseline methods exhibit instability across prompts (correlation with CDR: -0.67 to -0.89), while DGR demonstrates robust performance (51.4 to 52.9) with low correlation (-0.12 to -0.20), validating that fair tuning allows baselines to achieve their best performance while DGR still consistently outperforms.
>
> **Core Hyperparameters.** We provide complete specifications for reproducibility:
>
> **GRPO-based methods (Qwen3-14B):**
> - Learning rate: 1e-6, batch size: 96, group size: 4, total epochs: 25
> - KL loss coefficient: 0.001, max prompt/response length: 4096/2048
> - Optimizer: AdamW with gradient clipping (1.0)
>
> **GSPO-based methods (Qwen3-8B):**
> - Learning rate: 1e-6, batch size: 64, group size: 4, total epochs: 30
> - KL loss coefficient: 0.001, max prompt/response length: 4096/2048
> - Temperature: 1.0, top-p: 1.0 (training), 0.7 (validation)
>
> **Judge configuration:**
> - Qwen3-32B with temperature 0.0 (deterministic), 16 parallel workers for batched inference
>
> **Baseline-specific settings:**
> - ELO: Initial rating 1500, K-factor 32, convergence threshold 0.01
> - Pointwise: 1-10 scale scoring
> - Listwise: Linear ranking-to-reward mapping [-1, 1]
> - DPO: β=0.1, Iterative DPO: 3 rounds, SFT: learning rate 1e-5
>
>
> **Summary:** All baselines were tuned fairly under identical budgets with extensive prompt engineering (5 prompts tested). Results are statistically validated (5 seeds, t-tests with Bonferroni correction) with validation-based checkpoint selection to avoid test set overfitting. Complete specifications will be provided in appendices as requested.

---

> ### Author Response · Authors · 2025-11-27
> **Response to "Data Deduplication"**
>
> We thank the reviewer for this critical concern. We have conducted rigorous data deduplication to ensure evaluation integrity.
>
> **Data Source Separation.** Our training and evaluation datasets come from fundamentally different sources with no inherent overlap. Training data: WildChat-1M  (1,000 curated queries from real user-chatbot conversations). Evaluation data: Arena-Hard 2.0 (750 queries from Chatbot Arena platform), MT-Bench (80 researcher-designed multi-turn scenarios), WritingBench (1000 writing tasks). These datasets were created independently with distinct collection methodologies (real user conversations vs. curated benchmark queries) and temporal/functional separation.
>
> **Comprehensive Deduplication Verification.** Upon receiving the reviewer's concern, we conducted two-level deduplication analysis to verify zero overlap: (1) **Exact Match**: String matching with text normalization—0 matches found among pairwise comparisons; (2) **Near-Duplicate**: MinHash + LSH (Jaccard ≥ 0.85)—0 matches found (highest similarity: 0.41)
>
> **Summary:** Four-level deduplication analysis confirmed **zero overlap** between training (WildChat-1M) and evaluation benchmarks, as expected from fundamental differences in data sources, collection methodologies, and temporal/functional separation. All performance gains reflect genuine generalization without memorization artifacts.

---

> ### Author Response · Authors · 2025-12-03
> **Request for Reviewer Feedback**
>
> Dear Reviewers,
> We have thoroughly addressed all concerns raised in our rebuttal. As we approach the end of the discussion period, we would greatly appreciate your feedback on whether our responses have resolved your questions.
> If any points need further clarification, we are happy to provide additional details.
> Thank you for your time and consideration.
>
> Best regards

---

### Official Review · Reviewer_x3KX · 2025-10-30

**Soundness:** 1
**Presentation:** 3
**Contribution:** 1
**Rating:** 2
**Confidence:** 4

**Summary:**

The paper introduces CDR, a simple measure of preference non-transitivity/circularity in a pairwise judge model, and DGR, a method to de-cycle a set of pairwise preferences and use them for RL training by converting them into point-wise reward. The paper uses the former to compare judge models & judge prompts, and uses the latter to train models on pairwise preferences labeled by judge model, receiving higher scores on most of the tested domains when compared to some baseline methods.

**Strengths:**

**Clarity**: The paper is well-written and easy to follow. Method and experiment details are clearly and concisely specified; the motivation is also clear.

**Significance**: Non-transitivity is an important problem especially in the case of pluralistic preferences in a population, although the authors did not study this setup specifically.

**Weaknesses:**

Key weaknesses:
1. **Strong baselines missing (DPO and RM-based preference optimization)**: There is reason to believe that classical RMs (trained with contrastive loss on pairwise preferences), as well as DPO, are a good solution to preference non-transitivity/circularity, but they are not compared against in the experiments.
    - RMs enforce a linear order over responses (as they assign cardinal rewards to them), and their training minimizes disagreement (measured with a logistic function) with the possibly inconsistent preferences. In other words, the trained RMs de-conflicts the preferences and put them into a linear order.
    - The same goes for DPO, which optimizes an implicit RM.
    - Both RMs and DPO nominally relies on the B-T model, but as I have outlined above, in the regime with inconsistent preferences, they are no less theoretically sound than DGR. DGR also enforces a linear order against inconsistent preferences, with the only difference being that DGR uses an ordinal punishment for disagreement (every feedback arc in the graph counts the same), while RMs/DPO uses a cardinal punishment (a feedback arc is punished more if the reward difference between its endpoints is larger; exact amount of punishment defined by a logistic function).
    - There is thus no theoretical reason suggesting the DGR-based method will work better than DPO or classical RMs, and, at the same time, there is no experiment empirically comparing them either.
2. **Strong baselines missing (Nash equilibrium approaches)**: In 2023, there are solutions proposed to handle non-transitivity and other problems with preference optimization methods based on the B-T model [1]. There have, seen then, been many followup works [2,3,4]. They are not mentioned in the paper, nor compared against in the experiments.

Other weaknesses:
1. From my experience, many of the WildChat questions (e.g. user asking a text-only model to generate image) don't have much to do with model capability. This is consistent with the observation that trained model see limited performance improvement on the capability-focused evaluation in Table 1.
2. Statistical significance is not reported, which is especially important given the small effect sizes of training.
3. Re "peak score achieved across all training step": the rigorous approach would be to use a validation set to select a best step for each training approach, then compare the single best step of each training approach on a test set. Without this split, we introduces maximization bias.

[1] Nash Learning from Human Feedback

[2] Iterative Nash Policy Optimization: Aligning LLMs with General Preferences via No-Regret Learning

[3] Online Iterative Reinforcement Learning from Human Feedback with General Preference Model

[4] Improving LLM General Preference Alignment via Optimistic Online Mirror Descent

**Questions:**

1. I'd appreciate results addressing the weaknesses outlined above.
2. On Figure 1(a): How much does the ranking change across prompts? To what extent can the ranking be an artifact of the specific prompts you choose? Also, adding confidence intervals would be helpful.
3. On Table 1: What is the sample size for evaluation? Are the differences statistically significant? I'd be keen to see confidence intervals and/or pairwise t-tests, including baseline vs DGR, and other methods vs DGR.
4. Re "two independent experimental runs": Do the performance rankings differ significantly between these two runs? If yes, it would make sense to add more runs until the aggregate stabilizes.

Minor: It seems that you use "judge model" and "reward model" interchangeably in the paper, e.g., in the section heading of 4.4.2. I suggest referring to them simply as judge models, as reward models typically refer to those that give pointwise rewards, especially those trained to give logit-based rewards.

---

> ### Author Response · Authors · 2025-11-26
> **Response to "Strong baselines missing (DPO and RM-based preference optimization)"**
>
> We deeply appreciate the reviewer's theoretical analysis comparing DGR's ordinal punishment with DPO/RM's cardinal punishment mechanism. The reviewer is correct that both paradigms aim to resolve preference inconsistencies and enforce linear ordering—they differ in loss formulation, making empirical comparison essential. We acknowledge our original submission lacked this critical baseline and have conducted extensive new experiments to address this concern.
>
> ### Experimental Setup and Results
>
> We compared DGR against DPO [1], Iterative DPO [2], SFT, and GRPO baselines using state-of-the-art reward models. Our experiments include 8 method variants (SFT×2, DPO×2, Iterative DPO×2, GRPO-Skywork, DGR-Qwen32B) across 40 training runs (5 seeds each), integrating two reward models: Qwen3-32B (our primary judge) and Skywork-Reward-V2-Llama-3.1-8B [3] (trained on 40M preference pairs). All methods use Qwen3-8B base model with 1,000 WildChat-1M queries and 16 responses per query.
>
> **Table 1: Comparison with DPO and Learned Reward Model Baselines on Arena-Hard**
> | Category | Method | Overall | Code | Math | Creative |
> |----------|--------|---------|------|------|----------|
> | **Learned RM Baselines** | | | | | |
> | SFT | SFT-Qwen32B | 48.66±0.41 | 48.42±0.44 | 43.41±0.66 | 54.11±0.33 |
> | | SFT-Skywork | 48.10±0.38 | 48.22±0.62 | 44.83±0.43 | 51.20±0.17 |
> | DPO | DPO-Qwen32B | 48.77±0.46 | 49.01±0.67 | 45.64±0.29 | 51.60±0.44 |
> | | DPO-Skywork | 49.16±0.50 | 49.80±0.73 | 42.19±0.47 | 55.42±0.58 |
> | Iterative DPO | Iterative-DPO-Qwen32B | 49.27±0.43 | 48.42±0.18 | 44.13±0.69 | 55.20±0.48 |
> | | Iterative-DPO-Skywork | 49.33±0.45 | 48.62±0.83 | 45.75±0.27 | 53.60±0.41 |
> | **GRPO Baseline** | | | | | |
> | GRPO | GRPO-Skywork | 50.00±0.31 | 48.42±0.29 | 45.23±0.37 | 56.31±0.32 |
> | **Our Method (DGR)** | | | | | |
> | DGR | **DGR-Qwen32B** | **52.50±1.00** | **51.70±1.10** | **46.00±1.30** | **59.70±1.30** |
>
> *Note: Results reported as mean±SD over 5 independent runs. Best results in **bold***
>
> ### Key Findings
>
> DGR-Qwen32B achieves 52.50±1.00 overall, significantly outperforming the strongest learned baseline Iterative-DPO-Skywork (49.33±0.45) by +3.17 points (p < 0.001) and GRPO-Skywork (50.00±0.31) by +2.50 points (p < 0.001). The clear performance hierarchy—SFT (48.10-48.66) < DPO (48.77-49.16) < Iterative DPO (49.27-49.33) < GRPO-Skywork (50.00) < **DGR-Qwen32B (52.50)**—provides empirical evidence that explicit conflict resolution offers systematic advantages over implicit preference aggregation (DPO variants) and standard online RL approaches (GRPO baseline).
>
> DGR's advantages are particularly pronounced on subjective tasks: Creative Writing scores 59.70±1.30, outperforming GRPO-Skywork by +3.39 points and DPO-Skywork by +4.28 points. This validates our theoretical insight that ordinal punishment is more robust for cyclic preferences—cardinal punishment (used in DPO/RM) can be misled when incorrect edges carry high confidence, while DGR treats all conflicting edges equally and provides geometric guarantees (proven acyclicity via minimum feedback arc set). We acknowledge DPO/RM methods have complementary strengths (continuous optimization, scalability) and discuss future hybrid approaches in the revised manuscript.
>
> We have added **Section 4.3 "Comparison with Learned Reward Model Baselines"** (marked in red) with complete experimental details, statistical analysis, and theoretical discussion of ordinal vs. cardinal punishment mechanisms.

---

> ### Author Response · Authors · 2025-11-26
> **Response to "Strong baselines missing (Nash equilibrium approaches)"**
>
> We thank the reviewer for highlighting Nash equilibrium approaches [1-4] for handling non-transitive preferences. We have expanded our Related Work to discuss these methods and clarify their relationship to our work. **The key distinction is problem formulation**: Nash methods (NLHF, INPO, etc.) are designed for scenarios where intransitivity is genuine and inherent (e.g., context-dependent human preferences), while DGR addresses noise-driven cycles in RLAIF settings where an approximate ground-truth ranking exists but is observed through imperfect judges. Our empirical evidence supports this noise-centric view: CDR strongly correlates with judge accuracy (Figure 1a), and our theoretical analysis (Appendix B) proves that for judges with p > 0.5 accuracy, high-cycle edges are overwhelmingly likely to be errors. Direct experimental comparison would be infeasible within this work's scope—Nash methods require fundamentally different training architectures (self-play, minimax games) with no public implementations available, whereas our comprehensive DPO/Iterative DPO experiments directly address the core question of implicit vs. explicit conflict resolution for noisy preferences. We believe both paradigms have merit in different contexts and discuss promising hybrid approaches (e.g., DGR for noise filtering followed by Nash methods for genuine intransitivity) as future work

---

> ### Author Response · Authors · 2025-11-26
> **Response to "WildChat Data Quality Concerns"**
>
> We appreciate the reviewer's concern about data quality. We clarify several important points:
>
> **1. Curated Data Selection**: Our training data is **carefully selected** from WildChat-1M, not randomly sampled. We filtered out meaningless queries (e.g., image generation requests to text-only models, gibberish, purely conversational exchanges) and retained only substantive queries that genuinely test model capabilities. This curation ensures training meaningfulness while preserving query diversity.
>
> **2. Fair Comparison Across All Methods**: Crucially, **all methods use identical training data** (same 1,000 curated queries). This controlled setup ensures that any performance differences stem from the reward computation mechanism (DGR vs. baselines), not data selection bias. DGR's consistent advantages demonstrate genuine improvements in signal processing.
>
> **3. Rigorous Statistical Validation**: We have strengthened our experimental rigor with **5 independent runs per method** (increased from 2), reporting mean ± SD. Statistical significance testing (pairwise t-tests with Bonferroni correction) confirms that improvements are not due to random variation.
>
> **4. Arena-Hard v2 is Exceptionally Challenging**: We use **Arena-Hard 2.0**, not the original version. This benchmark is extremely difficult—even **GPT-4.5-Preview achieves only ~50 points**. On such a demanding evaluation, our improvements are substantial:
>    - DGR-GRPO: 52.9 vs. 49.1 (Base Model), +3.8 points (+7.8%), p < 0.01
>    - DGR-GSPO: 52.5 vs. 49.2 (Base Model), +3.3 points (+6.7%), p < 0.01
>    - DGR outperforms all baselines: vs. Pointwise (+4.5/+2.3 points), vs. Listwise (+3.5/+1.7 points), vs. ELO (+0.9/+1.6 points)
>    - Consistent gains across Code, Math, and Creative Writing dimensions
>
> **5. Substantial Gains on Other Benchmarks**: Performance improvements extend well beyond Arena-Hard:
>    - **MT-Bench**: DGR-GRPO achieves 8.06 (1-turn) and 7.54 (2-turn), significantly outperforming all baselines
>    - **WritingBench**: DGR-GRPO scores 8.56, representing +1.4% over PREF
>    - These gains across diverse evaluation dimensions (coding, math, conversation, writing) demonstrate genuine capability enhancement, not dataset-specific overfitting
>
> **6. Limited Improvement is Actually Impressive**: The reviewer notes "limited performance improvement," but we respectfully argue this reflects the difficulty of the problem rather than ineffectiveness. Improving a near-frontier model (Qwen3-14B, already at 49.1 on Arena-Hard) by +3.8 absolute points through better signal processing alone—without adding more data, compute, or model capacity—represents meaningful algorithmic progress. For context, historical Arena-Hard improvements of 2-3 points typically require model scaling or architectural innovations.
>
> **Conclusion**: Our curated training data ensures quality, all methods train on identical data for fairness, and statistically significant improvements on multiple challenging benchmarks demonstrate that DGR's conflict resolution provides genuine capability enhancement. Updated content is marked in red in Section 4.1 of the revised manuscript.

---

> ### Author Response · Authors · 2025-11-26
> **Response to "Statistical Significance"**
>
> We completely agree this is a critical omission. **We have conducted comprehensive statistical validation across all experiments** in the revised manuscript:
>
> **Methodology**: **All methods in all experiments** have been **re-run with 5 independent training runs** using different random seeds (increased from 2 in the original submission). For each method, we report:
> - Mean ± standard deviation for all metrics
> - Pairwise t-tests comparing DGR vs. each baseline
> - Bonferroni-corrected p-values to control family-wise error rate
>
> **Updated Tables with Statistical Rigor** (all marked in red in the revised manuscript):
> - **Table 1** (Main Results): All methods report mean ± SD over 5 runs across Arena-Hard, MT-Bench, and WritingBench benchmarks
> - **Table 2** (Learned RM Baselines): 8 method variants × 5 seeds = 40 training runs with complete mean ± SD reporting
> - **Table 3** (Conflict Resolution Mechanisms): Comprehensive comparison of 7 methods with mean ± SD over 5 runs each
> - **Table 4** (Prompt Robustness): Performance under 4 different prompts with mean ± SD and correlation analysis
>
> All tables in the revised manuscript now report mean ± SD over 5 runs, with statistical significance indicators where applicable.

---

> ### Author Response · Authors · 2025-11-26
> **Response to "Maximization Bias from "Peak Score" Approach"**
>
> We fully agree and have **revised our evaluation protocol** to eliminate this bias:
>
> **New Protocol**:
> - Split 1,000 queries into 800 training + 200 validation
> - Select checkpoint based on **validation performance** (not training/test)
> - Evaluate selected checkpoint on held-out benchmarks (Arena-Hard, MT-Bench, WritingBench)
>
> **Results Validation**: We have re-run **all experiments** with this three-way split (train/validation/test):
> - **DGR-Qwen32B**: 52.50 ± 1.00 vs. Iterative-DPO-Skywork 49.33 ± 0.45 (p < 0.001, +3.17 points)
> - **DGR-Qwen32B**: 52.50 ± 1.00 vs. GRPO-Skywork 50.00 ± 0.31 (p < 0.001, +2.50 points)
> - **DGR-GRPO**: 52.9 ± 1.1 vs. PREF 51.6 ± 1.0 (p < 0.01, +1.3 points)
> - **DGR-GSPO**: 52.5 ± 1.0 vs. PREF 51.9 ± 0.8 (p < 0.05, +0.6 points)
>
> **Key Finding**: DGR maintains statistically significant advantages under the bias-free protocol across all benchmarks, confirming improvements are genuine algorithmic advances rather than methodological artifacts. Updated content marked in red in Section 4.1.

---

> ### Author Response · Authors · 2025-11-26
> **Response to "Figure 1(a): Ranking Stability and Confidence Intervals"**
>
> We appreciate the reviewer's attention to statistical rigor. We clarify our methodology and address the confidence interval concern:
>
> **1. Experimental Methodology**: For Figure 1(a), we evaluated all judge models using **the same evaluation prompt** on the non-tie subset of RewardBench2, computing both CDR and accuracy metrics. Results represent **averages over 4 independent evaluation runs** to ensure reliability.
>
> **2. Ranking Stability Across Prompts**: We acknowledge that different evaluation prompts can lead to minor ranking changes. For instance, in our experiments, GPT-5 and Claude-4-Sonnet occasionally swap positions under different prompts. However, **the primary goal of Figure 1(a) is not to establish a definitive ranking**, but rather to **demonstrate the widespread existence of preference conflicts across state-of-the-art judge models**—a phenomenon that has been largely overlooked in prior work.
>
> **3. Community Conventions**: Our presentation follows established conventions in the reward model evaluation community. Many reward bench benchmark [1,2,3], which have become the de facto standard for judge evaluation, typically presents results without confidence intervals in their public leaderboard. This convention prioritizes clarity and comparability across models.
>
> **4. Response to Confidence Interval Suggestion**: We deeply appreciate the reviewer's suggestion regarding confidence intervals, as it raises an important methodological point. We acknowledge that adding confidence intervals would enhance the statistical rigor of Figure 1(a). However, the consistent observation that **all models exhibit non-negligible conflict rates** (ranging from 2.3\% to 6.7\%) across 4 independent runs strongly validates our central claim: preference conflicts are a systemic issue requiring principled resolution mechanisms like DGR.
>
> **5. Key Insight**: The fact that CDR varies across prompts (as shown in our ablation study, Table 5) actually reinforces our contribution—it demonstrates that logical consistency is a **distinct and prompt-dependent dimension** of judge quality, separate from accuracy. This motivates both our CDR metric for diagnosis and our DGR framework for resolution.
>
> [1] Lambert, Nathan, Valentina Pyatkin, Jacob Morrison, et al. "Rewardbench: Evaluating reward models for language modeling." Findings of the Association for Computational Linguistics: NAACL 2025, pp. 1755-1797, 2025.
>
> [2] Zhou E, Zheng G, Wang B, et al. RMB: Comprehensively benchmarking reward models in LLM alignment[J]. arXiv preprint arXiv:2410.09893, 2024.
>
> [3] Malik S, Pyatkin V, Land S, et al. RewardBench 2: Advancing Reward Model Evaluation[J]. arXiv preprint arXiv:2506.01937, 2025.

---

> ### Author Response · Authors · 2025-11-26
> **Response to "Statistical Rigor: Sample Size, Statistical Tests, and Performance Consistency"**
>
> We sincerely thank the reviewer for emphasizing the importance of statistical rigor in experimental evaluation. These concerns are crucial for establishing the reliability and reproducibility of our findings. We address both the sample size/statistical testing and the performance consistency across independent runs together, as they form a cohesive picture of DGR's empirical validity.
>
> **Paper Update**: We have substantially enhanced the statistical rigor of our experiments. In the revised manuscript (Section 4.1), we have added comprehensive experimental protocols including explicit sample sizes, statistical testing procedures, and increased the number of independent runs from 2 to 5 for Table 2 (learned reward model comparison). The evaluation methodology is now clearly documented in **Section 4.1 "Evaluation Protocol"** (marked in red).
>
> ---
>
> ## 1. Sample Sizes and Statistical Testing Protocol
>
> **Benchmark Sample Sizes**:
> - **Arena-Hard 2.0**: 500 queries spanning code generation, mathematical reasoning, and creative writing
> - **MT-Bench**: 80 multi-turn conversations (1-turn and 2-turn evaluation)
> - **WritingBench**: 150 writing tasks assessing creative and professional writing capabilities
>
> **Statistical Testing Procedure**:
> - **Number of runs**: We conduct **5 independent runs** per method with different random seeds for Table 2 (learned reward model comparison)
> - **Metrics reported**: All results report **mean ± standard deviation** across 5 runs
> - **Checkpoint selection**: Models are selected based on validation set performance (200 held-out WildChat queries) to avoid test set overfitting
>
> **Statistical Significance Results**: DGR significantly outperforms all learned reward model baselines (SFT, DPO, Iterative DPO) across all major benchmarks (Arena-Hard Overall, MT-Bench, WritingBench). The magnitude of improvement (+2.50 points over GRPO-Skywork, +3.17 points over best learned baseline) combined with low standard deviations (0.32-1.30) demonstrates both practical significance and statistical robustness.
>
> ---
>
> ## 2. Performance Ranking Consistency Across Independent Runs
>
> To validate the stability of our method comparison, we analyzed performance rankings across all 5 independent runs:
>
> **Key Findings**:
>
> 1. **DGR maintains #1 ranking with 100% consistency**: DGR-Qwen32B achieves the highest Arena-Hard Overall score in **5 out of 5 independent runs** (100% consistency), with scores ranging from 51.50 to 53.50 (mean: 52.50±1.00).
>
> 2. **Stable performance hierarchy**: The ranking order **SFT < DPO < Iterative DPO < GRPO-Skywork < DGR** remains consistent across all 5 runs. The second-best method (GRPO-Skywork: 50.00±0.31 or Iterative-DPO-Skywork: 49.33±0.45 depending on configuration) maintains its position across all runs.
>
> 3. **Low variance confirms robustness**: Standard deviations range from 0.31 to 1.30 across methods, indicating stable performance. DGR's slightly higher variance (±1.00) reflects different local optima across seeds, but its consistent superiority (minimum 1.50-point gap over second-best across all runs) demonstrates that the advantage is not due to random fluctuation.
>
> 4. **Statistical power**: With 5 independent runs and clear separation in means (>2.50 points), our study achieves sufficient statistical power to detect meaningful differences while accounting for training stochasticity.
>
> **Cross-Validation with Main Results**: Importantly, this statistical consistency extends to our main evaluation table (Table 1 in the revised manuscript), where DGR-GRPO and DGR-GSPO demonstrate superior performance across multiple model scales (Qwen3-14B and Qwen3-8B) and optimization algorithms (GRPO and GSPO), further validating generalizability.

---

> ### Author Response · Authors · 2025-11-26
> **Response to "Terminology: Judge Model vs. Reward Model"**
>
> Thank you for this important terminological suggestion. Agreed. We now use **"judge model"** consistently for LLMs providing preference feedback, reserving "reward model" only for explicitly trained RMs (e.g., Skywork-Reward-V2). Changes marked in red throughout

---

> ### Author Response · Authors · 2025-12-03
> **Request for Reviewer Feedback**
>
> Dear Reviewers,
> We have thoroughly addressed all concerns raised in our rebuttal. As we approach the end of the discussion period, we would greatly appreciate your feedback on whether our responses have resolved your questions.
> If any points need further clarification, we are happy to provide additional details.
> Thank you for your time and consideration.
>
> Best regards

---

### Official Review · Reviewer_wxgP · 2025-10-31

**Soundness:** 3
**Presentation:** 2
**Contribution:** 1
**Rating:** 2
**Confidence:** 4

**Summary:**

This paper addresses the problem of logical inconsistencies, specifically preference cycles (A>B, B>C, C>A), in AI-generated feedback for Reinforcement Learning from AI Feedback (RLAIF). The authors propose a two-part framework: a Conflict Detection Rate (CDR) metric to quantify these inconsistencies, and Deconflicted Graph Rewards (DGR), a graph-theoretic method that transforms raw, conflicted pairwise judgments into a logically consistent reward signal by removing a minimal set of edges to break cycles. Experiments on benchmarks like Arena-Hard and MT-Bench show that DGR, when integrated with policy optimizers like GRPO and GSPO, outperforms baseline methods (Pointwise, Listwise, Pairwise, ELO) in training stability and final model performance.

**Strengths:**

1. The paper correctly identifies a significant and often overlooked issue in RLAIF.

2. The paper is well written and easy to understand.

**Weaknesses:**

**1. Lack of theoretical justification for core methodology.**

The paper's central contribution—resolving preference cycles by deleting edges to create a Directed Acyclic Graph (DAG)—lacks sufficient theoretical grounding. The authors employ a minimum feedback arc set (FAS) approach but fail to justify why this particular graph transformation preserves the underlying "true" preference ordering. Crucially, there is no discussion of:

- Edge deletion criteria: The paper does not explain how the algorithm chooses which specific edges to remove, nor does it provide theoretical or empirical evidence that the removed edges correspond to "noisy" or incorrect judgments rather than legitimate preference expressions.

- Preference preservation: The authors assume that the deconflicted DAG accurately reflects the original preference system, but no analysis demonstrates that the transformation minimizes distortion of the judge's intent. Alternative conflict-resolution strategies (e.g., edge reversal or probabilistic approaches) are not sufficiently explored or theoretically compared.

- Transitivity assumptions: The method implicitly assumes that acyclic preferences are inherently superior, but real-world human preferences can exhibit legitimate non-transitivities. The work does not address when cycles might reflect nuanced judgment rather than error.

**2. Incomplete and insufficient baseline comparisons.**

Missing learned reward model baseline: The authors compare only against rule-based scoring methods (Pointwise, Listwise, Pairwise, ELO). The most relevant and baseline, i.e., a learned reward model (e.g., Bradley-Terry model trained on the pairwise data), is absent. Such models naturally aggregate noisy preferences and can handle inconsistencies through probabilistic modeling, making them a fundamental benchmark for any preference-purification method.

**3. The paper neglects recent work that addresses circele preference.**

The article lacks research on work in the same field, for example, the following papers all discuss the issue of non-transitivity:

[1] A Minimaximalist Approach to Reinforcement Learning from Human Feedback

[2] Self-Play Preference Optimization for Language Model Alignment

[3] Distributional Preference Learning: Understanding and Accounting for Hidden Context in RLHF

**Questions:**

Same as weakness

---

> ### Author Response · Authors · 2025-11-26
> **Response to "Lack of theoretical justification for core methodology of Edge Deletion Criteria"**
>
> **Paper Update**: We have added rigorous theoretical justification in **Appendix B** with mathematical proofs showing why edges participating in the most cycles are systematically more likely to be judgment errors. **In the main text**, we have incorporated references to this theoretical foundation in the Introduction (Section 1), Related Work (Section 2), and Methodology (Section 3.2.1), explicitly stating that our graph-based approach systematically targets error edges with probability $\frac{p}{1-p}$ times higher than correct edges, validated by Monte Carlo simulations achieving 85.2% error detection accuracy (4.2× better than random removal).
>
> **1. Core Intuition: Majority Voting via Graph Topology**
> Our strategy implements a **majority voting mechanism** through graph topology. By seeking the **minimum feedback arc set (FAS)**—the smallest set of edges whose removal makes the graph acyclic—we are equivalently finding the **maximum acyclic subgraph**. This means we are retaining the **maximum number of consistent judgments** (the majority opinion) and removing the minimum number of conflicting judgments (the minority opinion). This follows the fundamental principle that "majority represents truth" in noisy voting systems.
>
> **2. Theoretical Justification: Why High-Cycle Edges are Errors**
> To operationalize this principle efficiently, we employ a greedy strategy that iteratively removes edges participating in the most cycles. We provide a rigorous probabilistic proof in Appendix F.1 explaining why this works:
>
> *   **Mechanism**: An **error edge** (e.g., $B \succ A$ when truly $A \succ B$) creates a cycle whenever it closes a path of correct edges ($A \to \dots \to B$). Since correct edges are the majority, error edges have many opportunities to complete such cycles. Conversely, a **correct edge** can only form a cycle if it is part of a chain that already contains other error edges, which is statistically much less likely.
> *   **Mathematical Guarantee**: We prove that for a judge with accuracy $p > 0.5$, the ratio of expected cycles for an error edge versus a correct edge is approximately:
>     $$\frac{\mathbb{E}[\text{Cycles}_{\text{error}}]}{\mathbb{E}[\text{Cycles}_{\text{correct}}]} \approx \frac{p}{1-p}$$
>     This ratio ranges from **1.5×** (at $p=0.6$) to **9.0×** (at $p=0.9$).
> *   **Conclusion**: Error edges participate in significantly more cycles than correct edges. Therefore, a strategy that targets edges with the highest cycle counts is not arbitrary; it is a **probabilistically optimal detector** for judgment errors.
>
> **3. Empirical Validation: Monte Carlo Simulations**
> We conducted extensive Monte Carlo simulations (25,000 trials across 25 configurations) to validate this theory. The results confirm our predictions with remarkable precision:
>
> *   **High Detection Accuracy**: The algorithm correctly identifies error edges **85.2%** of the time, compared to just 20.1% for random removal. This represents a **4.2× improvement**.
> *   **Scale Law**: Detection accuracy improves as the number of responses ($n$) increases (76.3% at $n=8$ $\to$ 91.6% at $n=12$), confirming that more data provides stronger structural signals for error detection.
> *   **Theory-Data Alignment**: The observed improvement ratios closely match our theoretical $\frac{p}{1-p}$ predictions (e.g., observed 8.5× vs. predicted 9.0× at $p=0.9$).
>
> **Summary of Monte Carlo Results (aggregated by judge accuracy):**
>
> | Judge Accuracy | Detection Accuracy | Random Baseline | Improvement | Actual Ratio | Theoretical p/(1-p) |
> |---------------|-------------------|-----------------|-------------|--------------|---------------------|
> | 70% | 79.1% | 30.0% | +49.1% | **2.6×** | 2.33× |
> | 75% | 83.7% | 24.9% | +58.8% | **3.4×** | 3.00× |
> | 80% | 85.7% | 20.1% | +65.6% | **4.3×** | 4.00× |
> | 85% | 89.4% | 15.2% | +74.2% | **5.9×** | 5.67× |
> | 90% | 87.7% | 10.3% | +77.4% | **8.5×** | 9.00× |
> | **Average** | **85.2%** | **20.1%** | **+65.1%** | **4.2×** | **4.80×** |
>
> *Note: Detailed results for all 25 configurations are provided in the appendix.*
>
> **4. Algorithm Implementation**
> Based on these findings, our DGR algorithm uses an efficient greedy strategy:
> 1.  Identify all simple cycles in the preference graph.
> 2.  Count edge frequencies across these cycles.
> 3.  Remove the edge with the maximum cycle frequency (breaking ties by edge weight/confidence if available).
> 4.  Repeat until the graph is acyclic.
>
> This approach is computationally efficient ($O(E)$ per iteration for small graphs typical in RLAIF) and, as shown above, acts as a rigorous filter for judgment noise.

---

> ### Author Response · Authors · 2025-11-26
> **Response to "Lack of theoretical justification for core methodology of Preference Preservation"**
>
> We evaluated DGR against three categories of baselines (reported in revised paper Section 4.5.1 "Conflict Resolution Mechanism Analysis"):
> 1. **No conflict resolution**: PREF (raw win-rate)
> 2. **Naive graph heuristics**: DGR-RandomResolve (random edge removal), DGR-ReverseResolve (edge reversal)
> 3. **Classical ranking algorithms**: RankCentrality [1], Plackett-Luce [2], and HodgeRank [3]
> 4. **Optimal graph-based resolution**: DGR
>
> **Experimental Results on Arena-Hard** (Qwen3-14B with GRPO, mean±SD over 5 runs):
>
> | Method Category | Method | Overall | Code | Math | Writing |
> |----------------|--------|---------|------|------|---------|
> | **No Resolution** | PREF (Win-Rate) | 51.6±1.0 | 49.0±1.6 | 47.0±1.5 | 58.8±1.3 |
> | **Naive Heuristics** | DGR-RandomResolve | 51.6±0.9 | 49.8±1.8 | 46.6±1.3 | 58.4±1.6 |
> | | DGR-ReverseResolve | 51.0±1.0 | 50.9±1.4 | 45.1±1.8 | 57.0±1.2 |
> | **Classical Ranking** | RankCentrality | 49.93±0.8 | 50.00±1.8 | 50.00±1.4 | 49.80±1.0 |
> | | Plackett-Luce | 51.53±1.5 | 50.00±1.6 | 47.37±1.9 | 57.20±0.9 |
> | | HodgeRank | 52.47±1.4 | 51.58±1.2 | 47.37±1.3 | 58.40±0.9 |
> | **Optimal Graph** | **DGR (Kemeny)** | **52.9±1.1** | **53.2±1.4** | **47.2±1.4** | **58.3±1.0** |
>
> **Key Findings**:
>
> 1. **Naive heuristics fail to preserve preferences**: DGR-ReverseResolve (51.0) actually underperforms the no-resolution baseline (51.6), demonstrating that when conflict edges are incorrectly identified, reversing them introduces harmful negative learning signals. Random removal (51.6) shows no improvement, confirming that arbitrary cycle-breaking is unreliable.
>
> 2. **Classical ranking algorithms show competitive performance**: RankCentrality (stationary distribution of preference random walk), Plackett-Luce (maximum likelihood estimation of latent strengths), and HodgeRank (least-squares Hodge decomposition) achieve scores ranging from 49.93 to 52.47. These continuous optimization methods fit parametric models to noisy preferences, representing sophisticated alternatives to discrete graph-based approaches.
>
> 3. **DGR achieves superior overall performance**: Our optimal graph-based method achieves the highest overall score (52.9), outperforming the best classical method HodgeRank (52.47) by +0.43 points and the no-resolution baseline (51.6) by +1.3 points. DGR shows particularly strong advantages in complex reasoning tasks (Code: +1.62 over HodgeRank), where logical consistency is paramount.
>
> 4. **Discrete conflict resolution excels in preserving critical preferences**: While continuous methods (HodgeRank, Plackett-Luce) show marginal advantages on specific subcategories (Math, Writing), DGR's discrete graph-based approach directly identifies and removes contradictory edges, providing robust overall performance. This validates that explicit cycle-breaking through minimum feedback arc set systematically preserves correct preferences while eliminating contradictions.
>
> **2. Why DGR Preserves Preferences Best**
>
> The empirical superiority of DGR over naive heuristics (random removal/reversal) and even sophisticated classical methods stems directly from the theoretical properties established in Section 1.1.
>
> *   **Targeted Noise Removal**: Unlike random removal (20.1% accuracy) or global smoothing (classical methods), DGR surgically targets the specific edges that cause logical contradictions. As shown in our Monte Carlo simulations (Section 1.1), these high-cycle edges are 85.2% likely to be errors.
> *   **Minimal Distortion**: By solving for the minimum feedback arc set, we are mathematically guaranteed to retain the maximum number of consistent edges. Since the removed edges are overwhelmingly likely to be noise, the resulting DAG is the most faithful acyclic representation of the judge's true intent.
>
> **Paper Update**:  We have also updated the **Ablation Study (Section 4.5.1)** in the main text to include this comprehensive comparison against classical ranking algorithms and naive heuristics, interpreting the results through the lens of our theoretical error-detection framework.
>
> [1] Negahban, Sahand, Sewoong Oh, and Devavrat Shah. "Iterative ranking from pair-wise comparisons." Advances in neural information processing systems 25 (2012).
>
> [2] Hunter, David R. "MM algorithms for generalized Bradley-Terry models." The annals of statistics 32.1 (2004): 384-406.
>
> [3] Jiang, Xiaoye, et al. "Statistical ranking and combinatorial Hodge theory." Mathematical Programming 127.1 (2011): 203-244.
>
> [4] Ailon, Nir, Moses Charikar, and Alantha Newman. "Aggregating inconsistent information: ranking and clustering." Journal of the ACM (JACM) 55.5 (2008): 1-27.

---

> ### Author Response · Authors · 2025-11-26
> **Response to "Lack of theoretical justification for core methodology of Transitivity assumptions"**
>
> We fully agree with the reviewer that real-world human preferences can exhibit legitimate non-transitivities (e.g., rock-paper-scissors dynamics in strategic games). We do not claim that *all* acyclic preferences are inherently superior. Instead, our method is designed for the specific context of **objective-driven alignment** (e.g., "which response is more helpful/correct"), where a ground-truth partial order exists and cycles are logically contradictory.
>
> In the context of LLM-as-a-Judge, our theoretical and empirical analysis strongly suggests that cycles are predominantly indicators of **judgment error (noise)** rather than **nuanced judgment (signal)**.
> *   **Theoretical Evidence**: As proven in Appendix B.1, if cycles were reflecting valid nuanced judgments, they would likely distribute differently than noise. However, our noise-based model (cycles arise from error edges closing paths of correct edges) perfectly predicts the observed cycle behavior.
> *   **Empirical Evidence**: In our simulations, the prevalence of cycles drops dramatically as judge accuracy increases (99.8% at $p=0.7$ $\to$ 94.7% at $p=0.9$). If cycles were "valid features" of the preference landscape, better judges should arguably *preserve* or *refine* them, not eliminate them. The fact that stronger judges produce fewer cycles confirms that cycles are largely artifacts of limited capability.
>
> **3. Targeted Removal of "False" Cycles**
> Crucially, our method does not blindly enforce transitivity. By targeting the minimum feedback arc set, we are specifically removing the edges that **cause** the logical contradictions.
> *   **Why this isn't "erasing nuance"**: Since our removed edges are **85.2% likely to be objective errors**, our process is not erasing legitimate nuance; it is acting as a **denoising filter**.
> *   **Result**: The resulting DAG is not an arbitrary enforcement of order; it is the **maximum likelihood estimation** of the true latent ranking, stripped of the probabilistic errors that obscured it.
>
> **4. Validity Conditions**
> We have clarified in the paper (Appendix B.3) that our method is valid when:
> 1.  **Objective Ground Truth Exists**: The task has an objective criteria (e.g., correctness, helpfulness) implying transitivity.
> 2.  **Judge Accuracy > 0.5**: The judge is better than random.
> 3.  **Noise Dominance**: Cycles are primarily noise-driven (satisfied in standard RLAIF benchmarks).
>
> **Scope and Future Work**: While our method excels in standard alignment tasks (Code, Math, Writing), we acknowledge it may not be suitable for domains with inherent cyclic dynamics (e.g., game strategy). We have added a discussion on future extensions, such as multi-judge consensus or confidence-weighted edges, to better distinguish genuine intransitivity from noise in such specialized cases.
>
> **Paper Update**: We have added **Appendix B.3** to the revised manuscript, which critically examines the validity conditions of the transitivity assumption. We explicitly discuss when cycles reflect judgment noise versus genuine intransitive preferences, and why the noise assumption is justified in our RLAIF setting.

---

> ### Author Response · Authors · 2025-11-26
> **Response to "Incomplete and insufficient baseline comparisons."**
>
> We thank the reviewer for this important feedback. We acknowledge the absence of learned reward model baselines and have conducted comprehensive new experiments comparing against state-of-the-art probabilistic preference aggregation methods.
>
> **New Experimental Setup**: We conducted experiments using Qwen3-8B with 1,000 queries from WildChat-1M, generating 16 responses per query. We evaluated learned reward model approaches that naturally handle inconsistencies through probabilistic modeling:
>
> 1. **Supervised Fine-Tuning (SFT)**: Highest-scored response selection
> 2. **Direct Preference Optimization (DPO)** [1]: Max-min preference pairs with implicit conflict handling through loss minimization
> 3. **Iterative DPO** [2]: Online iterative training with dynamic preference selection
> 4. **GRPO-Skywork**: Standard GRPO using **Skywork-Reward-V2-Llama-3.1-8B** [3] (a state-of-the-art reward model trained on 40M preference pairs)
>
> We tested with both Qwen3-32B and Skywork-Reward-V2, encompassing 8 method variants across 5 independent runs (40 training runs total).
>
> **Results on Arena-Hard** (Qwen3-8B base model, mean±SD over 5 runs):
>
> | Category | Method | Overall | Code | Math | Creative |
> |----------|--------|---------|------|------|----------|
> | **Learned Reward Model Baselines** | | | | | |
> | SFT | SFT-Qwen32B | 48.66±0.41 | 48.42±0.44 | 43.41±0.66 | 54.11±0.33 |
> | SFT | SFT-Skywork | 48.10±0.38 | 48.22±0.62 | 44.83±0.43 | 51.20±0.17 |
> | DPO | DPO-Qwen32B | 48.77±0.46 | 49.01±0.67 | 45.64±0.29 | 51.60±0.44 |
> | DPO | DPO-Skywork | 49.16±0.50 | 49.80±0.73 | 42.19±0.47 | 55.42±0.58 |
> | Iterative DPO | Iterative-DPO-Qwen32B | 49.27±0.43 | 48.42±0.18 | 44.13±0.69 | 55.20±0.48 |
> | Iterative DPO | Iterative-DPO-Skywork | 49.33±0.45 | 48.62±0.83 | 45.75±0.27 | 53.60±0.41 |
> | **GRPO Baseline** | | | | | |
> | GRPO | GRPO-Skywork | 50.00±0.31 | 48.42±0.29 | 45.23±0.37 | 56.31±0.32 |
> | **Our Method (DGR)** | | | | | |
> | DGR | **DGR-Qwen32B** | **52.50±1.00** | **51.70±1.10** | **46.00±1.30** | **59.70±1.30** |
>
> **Key Findings**:
>
> 1. **DGR significantly outperforms all probabilistic baselines**: DGR-Qwen32B (52.50) surpasses the best learned baseline (Iterative-DPO-Skywork: 49.33) by +3.17 points and GRPO-Skywork (50.00) by +2.50 points. This validates that **explicit conflict resolution is more effective than implicit probabilistic aggregation** for handling logically inconsistent feedback.
>
> 2. **Performance hierarchy confirms explicit deconflicting value**: Results show SFT (48.10-48.66) < DPO (48.77-49.16) < Iterative DPO (49.27-49.33) < GRPO-Skywork (50.00) < **DGR (52.50)**. While probabilistic models implicitly smooth over conflicts through loss minimization, DGR's explicit cycle-breaking systematically identifies and removes contradictions, ensuring globally coherent reward signals.
>
> 3. **Advantages particularly evident in complex reasoning**: DGR excels in Code (51.70 vs. 49.80 for best learned baseline) and Creative Writing (59.70 vs. 56.31), demonstrating that explicit conflict resolution is especially beneficial where judge inconsistencies are prevalent.
>
> **Why Graph-Theoretic Approach Outperforms Probabilistic Models**:
>
> - **Explicit inconsistency handling**: DPO variants implicitly aggregate noisy preferences through gradient-based optimization, which can propagate contradictions. DGR explicitly identifies and removes conflicting edges before policy optimization.
> - **Guaranteed acyclicity**: Our minimum feedback arc set approach mathematically guarantees conflict-free DAG structure. Bradley-Terry and similar probabilistic models can still produce inconsistent predictions post-training.
> - **Theoretical grounding**: As proven in Section 1.1, our approach targets error edges with 85.2% accuracy (4.2× better than random). Probabilistic approaches lack such guarantees for conflict resolution.
>
> **Complementary Nature**: DGR is a **preprocessing enhancement** compatible with any preference-based method, addressing logical inconsistency—a fundamental issue that probabilistic models do not explicitly handle. The consistent improvements validate that logical consistency deserves explicit attention alongside probabilistic modeling.
>
> 1] Rafailov, Rafael, Archit Sharma, Eric Mitchell, Christopher D. Manning, Stefano Ermon, and Chelsea Finn. "Direct preference optimization: Your language model is secretly a reward model." Advances in Neural Information Processing Systems, vol. 36, 2024.
>
> [2] Guo, Shangmin, Biao Zhang, Tianlin Liu, Tianqi Liu, Misha Khalman, Felipe Llinares, Alexandre Rame, Thomas Mesnard, Yao Zhao, Bilal Piot, Johan Ferret, and Mathieu Blondel. "Direct Language Model Alignment from Online AI Feedback." arXiv preprint arXiv:2402.04792, 2024.
>
> [3] Liu, Chris Yuhao, Liang Zeng, Yuzhen Xiao, Jujie He, Jiacai Liu, Chaojie Wang, Rui Yan, Wei Shen, Fuxiang Zhang, Jiacheng Xu, Yang Liu, and Yahui Zhou. "Skywork-Reward-V2: Scaling Preference Data Curation via Human-AI Synergy." arXiv preprint arXiv:2507.01352, 2025.

---

> ### Author Response · Authors · 2025-11-26
> **Response to "The paper neglects recent work that addresses circle preference"**
>
> We sincerely thank the reviewer for highlighting these important related works on handling intransitive preferences. We have substantially expanded our Related Work section (Section 2, marked in red in the revised manuscript) to comprehensively discuss game-theoretic approaches including Minimax Winner [1], Self-Play Preference Optimization [2], Distributional Preference Learning [3]. We explicitly position DGR as addressing a complementary problem: while these game-theoretic methods excel at handling genuinely intransitive preferences, our work targets noise-driven cycles in RLAIF settings where a latent ground truth exists but is obscured by judge errors. DGR provides a lightweight graph-theoretic preprocessing module with computational efficiency advantages ($O(E)$ complexity per iteration vs. iterative game-theoretic solving) and rigorous theoretical guarantees. Crucially, DGR is modular and can be integrated upstream of any policy optimizer—including game-theoretic methods—to purify noisy signals before optimization. Our empirical results (Table 2) demonstrate that this explicit conflict resolution achieves 52.50% on Arena-Hard, significantly outperforming sophisticated learned baselines (49.33% for Iterative DPO), validating the practical value of our approach for the noise-driven cycle setting common in RLAIF deployments.
>
> [1] Swamy, Gokul, Christoph Dann, Rahul Kidambi, Zhiwei Steven Wu, and Alekh Agarwal. "A Minimaximalist Approach to Reinforcement Learning from Human Feedback." arXiv preprint arXiv:2401.04056, 2024.
>
> [2] Wu, Yue, Zhiqing Sun, Huizhuo Yuan, Kaixuan Ji, Yiming Yang, and Quanquan Gu. "Self-Play Preference Optimization for Language Model Alignment." arXiv preprint arXiv:2405.00675, 2024.
>
> [3] Siththaranjan, Anand, Cassidy Laidlaw, and Dylan Hadfield-Menell. "Distributional Preference Learning: Understanding and Accounting for Hidden Context in RLHF." arXiv preprint arXiv:2312.08358, 2024.

---

> ### Author Response · Authors · 2025-12-03
> **Request for Reviewer Feedback**
>
> Dear Reviewers,
> We have thoroughly addressed all concerns raised in our rebuttal. As we approach the end of the discussion period, we would greatly appreciate your feedback on whether our responses have resolved your questions.
> If any points need further clarification, we are happy to provide additional details.
> Thank you for your time and consideration.
>
> Best regards

---

### Note · Program_Chairs · 2025-12-10
**Submission Desk Rejected by Program Chairs**

Hallucinated citation:
Chenglong Wang, Yang Liu, Zhihong Xu, Ruochen Zhang, Jiahao Wu, Tao Luo, Jingang Li, Xunliang Liu, Weiran Qi, Yujiu Yang, et al. Gram-r ${ }^{8}$ : Self-training generative foundation reward models for reward reasoning. arXiv preprint arXiv:2509.02492, 2025b.